# Development-Related miRNA Expression and Target Regulation during Staggered In Vitro Plant Regeneration of Tuxpeño VS-535 Maize Cultivar

**DOI:** 10.3390/ijms20092079

**Published:** 2019-04-27

**Authors:** Brenda A. López-Ruiz, Vasti T. Juárez-González, Estela Sandoval-Zapotitla, Tzvetanka D. Dinkova

**Affiliations:** 1Departamento de Bioquímica, Facultad de Química, Universidad Nacional Autónoma de México, 04510 Ciudad de México, CDMX, Mexico; ana_bell_89@hotmail.com (B.A.L.-R.); vasti.juarez.gonzalez@gmail.com (V.T.J.-G.); 2Jardín Botánico, Instituto de Biología, Universidad Nacional Autónoma de México, 04510 Ciudad de México, CDMX, Mexico; esz@ib.unam.mx

**Keywords:** callus, de novo shoot regeneration, maize, microscopy, miRNA regulation

## Abstract

In vitro plant regeneration addresses basic questions of molecular reprogramming in the absence of embryonic positional cues. The process is highly dependent on the genotype and explant characteristics. However, the regulatory mechanisms operating during organ differentiation from in vitro cultures remain largely unknown. Recently, miRNAs have emerged as key regulators during embryogenic callus induction, plant differentiation, auxin responses and totipotency. Here, we explored how development-related miRNA switches the impact on their target regulation depending on physiological and molecular events taking place during maize Tuxpeño VS-535 in vitro plant regeneration. Three callus types with distinctive regeneration potential were characterized by microscopy and histological preparations. The embryogenic calli (EC) showed higher miRNA levels than non-embryogenic tissues (NEC). An inverse correlation for miR160 and miR166 targets was found during EC callus induction, whereas miR156, miR164 and miR394 displayed similar to their targets RNA accumulation levels. Most miRNA accumulation switches took place early at regenerative spots coincident with shoot apical meristem (SAM) establishment, whereas miR156, miR160 and miR166 increased at further differentiation stages. Our data uncover particular miRNA-mediated regulation operating for maize embryogenic tissues, supporting their regulatory role in early SAM establishment and basipetala growth during the in vitro regeneration process.

## 1. Introduction

In vitro plant regeneration is usually achieved through somatic embryogenesis (SE) or de novo shoot organogenesis [1]. Somatic embryo induction represents full expression of totipotency, while shoot regeneration reflects cell pluripotency [2]. Both regeneration pathways are dependent on phytohormone perception, cell division and dedifferentiation to acquire organogenic competence, organ initiation and development. SE and de novo shoot organogenesis are broadly used, either for basic research or practical applications. However, in contrast to the embryonic pathway, the de novo shoot organogenic pathway is more valued in biotechnological breeding methods because the plant explants and the in vitro conditions are comparatively simpler and more robust [3].

For maize, plant regeneration was early described using immature embryos as explants to produce embryogenic calli (EC) type II and subsequently somatic embryos [4]. Initially, high EC frequency was restricted to few inbred lines, such as Hi-II, resulting from A188 and B73 crosses, whereas other genotypes produced mostly non-embryogenic callus (NEC) [5,6]. Further research reported in vitro maize regeneration via organogenesis [7,8]. Recently, other genotypes cultured in different geographic regions have proven to efficiently undergo SE and plant regeneration. Such is the case of a Chinese inbred elite line, 18-599R [9] and the Mexican improved variety VS-535 developed by Instituto Nacional de Investigaciones Forestales, Agricolas y Pecuarias (INIFAP) from the Tuxpeño landrace genotype [10,11]. The culture medium, photoperiod, phytohormone ratio and other conditions of the in vitro culture can be adjusted to promote either SE or organogenesis from the induced EC [12].

The molecular pathway to shoot regeneration primarily involves auxin-, cytokinin- and Shoot Apical Meristem (SAM)-related genes, including WUSCHEL (WUS), SHOOT MERISTEMLESS (STM) and CUP-SHAPED COTYLEDON 2 (CUC2) [3,13]. Their expression appears regulated at transcriptional and post-transcriptional levels. Many of the corresponding transcripts are targeted by microRNAs (miRNAs), which have been shown to act as mobile signals to restrict gene expression at particular plant tissues [14,15].

miRNAs are small (20–22 nt) non-coding RNAs that have shaped as master regulatory molecules for specific target mRNAs based on sequence complementarity. In *Arabidopsis thaliana*, identification of miRNA targets and the availability of mutants in the miRNA biogenesis pathway have evidenced their relevance in plant development [16,17,18]. Genome-wide studies have revealed major miRNA changes during EC induction in Arabidopsis [19], rice [20], wheat [21], maize [22] and many other plant species. In maize, particular miRNA levels do not only change during dedifferentiation, but also during the establishment of EC proliferation. A follow-up on development-related (miR156, miR164) and stress-related (miR398, miR408, miR528) miRNAs, as well as on their targets, showed contrasting patterns during VS-535 plant regeneration depending on hormone presence and light [23].

According to several recent reports, miR156, miR160, miR164, miR165/166 and miR394 are relevant for SAM development because they show specific spatio-temporal localization to inhibit mRNAs corresponding to transcription factors: SQUAMOSA PROMOTER BINDING-LIKE (SPL) 10/11, CUC1/2, AUXIN RESPONSE FACTOR (ARF) 10/16/17, HOMEODOMAIN-LEUCINE ZIPPER (HD-ZIP) III and LEAF CURLING RESPONSIVENESS (LCR), respectively [14]. The overexpression of miR165a leads to a drastic reduction on HD-ZIP III transcript levels and failure of SAM formation [24]. It has been suggested that balanced regulation of miRNA165/166 abundance is essential for HD-ZIP III-mediated spatio-temporal induction of WUS expression and de novo SAM initiation [25]. Moreover, miR156 and its target SPL9 were related to age-dependent developmental decline of shoot regenerative capacity in *Arabidopsis thaliana* and *Nicotiana tabacum*. Zhang et al. found that a reduction in miR156 levels and gradual increase of SPL9 lead shoot regenerative capacity by attenuating the cytokinin signal pathway [18]. In addition, it was found that miR160a negatively regulates shoot regenerative capacity through down-regulation of ARF10 and is more abundant in callus tissues unable to undergo de novo SAM initiation compared to those with shoot regenerative capacity [26].

Here we aimed to integrate miRNA-mediated regulation of key targets during in vitro maize plant regeneration for the Tuxpeño VS-535 cultivar using a previously established method of callus induction, proliferation and differentiation. To associate particular miRNA fates with regeneration capacity we compared the original explant (immature embryo, IE) with de-differentiated tissues displaying contrasting morphological and regenerative characteristics (EC and NEC), as well as during four reproducible phases of plant regeneration. According to our results, SAM formation-related miRNAs and their targets are significantly more abundant in EC than NEC, whereas WUS levels are similar in these tissues and higher than in the original explant. The regulation mediated by miR156, miR160 and miR166 nodes operates at early leaf differentiation, correlating with de novo SAM formation and reproducibly precedes root appearance. On the other hand, miR164 and miR394 decrease or relocate to specific regions in early regenerative spots and increase at later regeneration stages. According to this, we conclude that the initial miRNA and target levels in the callus underlie its regenerative potential and guide its ability to promote shoot appearance during plant regeneration from maize VS-535 in vitro cultures.

## 2. Results

### 2.1. Morphology of Dedifferentiated Tissues

According to previous reports, maize EC induction requires immature embryos (IE) at 15–18 days after pollination (DAP) as explant to obtain greater proportion of embryogenic dedifferentiated tissues, particularly EC Type II [6,11,27]. However, callus morphology might be heterogeneous, depending on the maize genotype. Tuxpeño VS-535 IEs collected at 15 DAP were 3–4 mm long and characterized by the suspensor absence (Figure 1; panels A and B). Upon induction, root protruded from the embryo and two kinds of cell masses became evident: one with waterish appearance at the side of the emerged root, in opposite orientation; the other localized at the scutellum side with clearly distinctive characteristics (Figure 1). Observation with scanning electron microscopy (SEM) showed a cumulus of smooth callus formed by small cells arising from the scutellum side (Figure 1E,F, arrowheads), whereas the one originating from the embryonic axis side, mostly from the root apical meristem (RAM), was characterized by the presence of enlarged cells (Figure 1E,F, arrows).

Upon successive subcultures of the dedifferentiated tissues, three callus types were easily distinguished: embryogenic callus (EC), yellow-soft non-embryogenic callus (Y-NEC) and white-compact non-embryogenic callus (W-NEC). All three calli types were tested for plant regeneration, but only the EC was able to develop plantlets. The EC was translucent, compact and friable (Figure 2A). These features were conserved through subsequent subcultures for over two years. According to the induction follow-up, the EC was generated on the upper (abaxial) surface of the scutellum placed in direct contact with the induction medium. Hence, it was likely scutellar-, epithelial-derived (Figure 1; panels E–G). The Y-NEC with watery yellowish to brownish appearance was able to massively proliferate during subculture passages (Figure 2D). It is possibly originated from coleorhiza (Figure 1; panels E, F and H) and was prone to form roots through organogenesis in the presence of light (Appendix A; panels C,D). The W-NEC showed an opaque smooth appearance and contrary to the EC was not friable (Figure 2G). It underwent aberrant development during plant regeneration (Appendix A; panel E). Both NEC types coexist with the EC during callus subcultures (Figure 2; panels J,K).

A morphologically distinct appearance for each callus type was evident by SEM analysis (Figure 2; panels B, E and H). The EC displayed small, isodiametric cells, while the Y-NEC consisted of large cells (200 µm), contributing to a rough surface appearance of the tissue. Histological analysis suggested the presence of meristematic zones in EC characterized by regular-shaped cells and dense nuclei (Figure 2C). As a distinctive feature, the W-NEC presented particular enrichment of tracheid-like structures in a disorganized fashion (Figure 2I). During callus subcultures, the W-NEC was discarded, whereas the other two callus types remained and were separated during the plant regeneration process.

### 2.2. Plant Regeneration from Tuxpeño VS-535 EC

Plant regeneration consisted in staggered hormone depletion and exposure to a photoperiod. The first visual evidence of regenerating tissues was the location of green spots, approximately 0.3 mm wide, on the callus surface, as previously reported [11,28]. After two weeks on proliferation medium (N6P) with half-concentration hormones and photoperiod, the green regenerative spots resembled protrusions corresponding a leaf tip (Figure 3A). These were designed as the 1st stage of development in our study. SEM and histological analyses revealed dense small meristematic cells at the apical region of each spot (Figure 3; panels A–C, Appendix A; panels A,B). Immediately under the regenerative spot, disorganized tracheid structures were observed (Appendix A; panel C, arrow).

After two additional weeks on N6P without hormones, considered as the 2nd stage of development, the regenerating tissue enlarged (1 mm of length) and acquired a subtle rolled leaf shape. Leaf primordium-like structures were covered by hairs and resembled epidermal cell type on the surface (Figure 3; panels D–F). Histology indicated high cell proliferation and presence of parenchyma throughout the tissue. More tracheid structures were visible immediately below the regenerating tissues (Appendix A; panels D–F). This structure developed in a typical leaf shape upon two further weeks on MS medium and was designed as the 3^rd^ stage of plant regeneration (Figure 3; G–J). At this stage, no evidence of root formation was detected. The leaf structure reached approximately 6 mm length and presented parallel veins characteristic of monocots. For the first time the tracheid structures appeared aligned and other cell types, such as chlorenchyma and amyloplasts-like, were readily distinguished (Appendix A; panels G–I).

Whole plantlet development was initiated after a subsequent passage of the regenerative tissues on MS. At this last stage, emerging roots and several fully developed leaves were observed (Figure 3; panels K–N). Root structures resembled more crown roots than a primary root, and arose at lateral sides of the stem. The histological analysis showed a typical RAM anatomy for the roots, as well as dorso-ventral polarity and well-formed vascular bundles for the leaves. However, a cross section of a plantlet stem shows no evidence of central vascular bundles. (Appendix A; panels J–N).

Under the above conditions, plant regeneration took about 42 days. It is important to strike that all stages were reproducibly observed at the indicated times for at least four independent regeneration batches. The number of regenerative spots per gram of callus is available in the (Appendix A; panel A,B). Interestingly, when root structures appeared before the leaf on callus, plant regeneration was not observed.

### 2.3. miRNA and Target Regulation in Tissues with Different Embryogenic Potential

We analyzed the abundances of miR156, miR160, miR164, miR166 and miR394, previously described as crucial factors in plant development [14]. The established Y-NEC (12 months of subculture) presented significantly decreased levels for all miRNAs, except for miR394, with respect to the original IE explant (Figure 4). However, only miR160 and miR166 were significantly reduced, whereas miR394 was highly accumulated in EC with respect to the explant. Particularly interesting observation was the higher abundance of miR156, miR164 and miR394 in EC than in Y-NEC suggesting a potential role regarding the embryogenic potential of dedifferentiated tissues (Figure 2).

All miRNA targets analyzed here have been experimentally confirmed through degradome data [29,30]. We found a noteworthy higher accumulation of miRNA targets in EC than in Y-NEC and IE (Figure 4; lower panels), except for SBP23 targeted by miR156, which was more accumulated in IE. According to this, for most miRNA targets there was no inverse correlation with the corresponding miRNA, particularly in Y-NEC. However, such correlation exists for miR160 and its targets AUXIN RESPONSE FACTORS (ARFs) 17 and 19, as well as for miR166 and RLD1/HD-ZIP III in comparisons between IE and EC. These results support central molecular dissimilarities between calli with different embryogenic potential, which might correlate with the regenerative capacity of a tissue. It is interesting to notice that the SE marker WUS2 showed greater abundance in both dedifferentiated tissues than in IE, supporting the high proliferation status of established calli. Nevertheless, an important feedback loop on WUS2 through miR394 and its target F-box (LCR) seems to be operational only for EC.

### 2.4. Embryogenesis-Related miRNA and Target Regulation During Plant Regeneration

At the 1st stage of plant regeneration from EC (two weeks on half-hormone depletion and photoperiod), a decrease was observed for most analyzed miRNAs although it was significant only for miR164 and miR394 (Figure 5; upper panels, lane b). During further regeneration stages we distinguished a burst in miRNA abundance at either the 2nd or the 3rd stage. While miR156, miR160 and miR166 increased at the 2nd stage, miR164 and miR394 did so at the 3rd stage (Figure 5; upper graphs, lanes c and d). Such behavior supports the notion that particular miRNAs operate in a development-dependent fashion during plant regeneration. According to the previously described morphology at each stage, embryogenesis-related miRNAs play a major role in leaf tip differentiation during VS-535 in vitro plant regeneration.

The highest levels of miR160 and miR166 were observed at the 2nd stage (six fold above EC), coincident with important reduction of ARF17 and RLD1 corresponding targets (Figure 5, lower panel). On the other hand, accumulation of miR156, miR64 and miR394 was appreciated at the 3rd stage of regeneration (four, two and three-fold above EC), but it was not mirrored by the reduction of their corresponding targets. On the contrary, an increase of SBP23 and CUC2, targeted by miR156 and miR164 respectively, was concomitantly found at this stage. As mentioned before, a negative correlation could not always be observed between a miRNA and its target. This could be due to a different regulatory mechanism (i.e., translation) or more likely, the inability to discern the miRNA action at particular cell types within the analyzed tissues.

A significant decrease in miR160 and miR166 levels was evident at the 3^rd^ developmental stage when the regenerative structures acquired fully developed leaf shape. In fully regenerated plantlets, miR156, miR160, miR164 and miR394 levels were significantly down-regulated, whereas miR166 appeared again up-regulated if compared to the preceding stage (Figure 5; lane e). According to our physiological observations, the miR166 abundance pattern might be related to root formation, as well as to the development of new leaves (Figure 3K–N).

During the early stages of plant regeneration, targets of miR156, miR160 and miR166 showed strong inverse correlation patterns with the corresponding miRNAs. On the other hand, CUC2 and F-Box/LCR, targeted by miR164 and miR394 respectively, did not show inverse correlation with the miRNA at any regeneration stage. Interestingly, it was possible to observe contrasting behavior for ARF17 and ARF19, targeted by the same miRNA, at later regeneration stages. This further supports the possibility of target-specific regulatory mechanisms depending on development or spatial restriction of the miRNA action.

Most of the targets analyzed here correspond to transcription factors (TF) that regulate gene expression in response to developmental signals in particular tissues. Maize SBP23 is orthologous to Arabidopsis SPL10/11 family of TF involved in vegetative and reproductive plant transitions [16]. CUC2 is associated with the establishment of embryo apical-basal polarity, stem cell identity and boundary specification between SAM and lateral organ formation [3,13]. RLD1 is an HD-ZIP III TF crucial for vascular patterning and SAM development [24]. Taking advantage of previously constructed gene regulatory networks based on maize RNA-seq expression data [31] we performed a search for miRNA targets as “regulators” in four tissues: seed, SAM, leaf and root (Appendix A). The results indicated that SBP23 and RLD1 had more interactions in leaf, CUC2 in SAM and ARF 17 in root. All four TF presented relevant interactions in the SAM tissue, playing a crucial role during maize in vitro plant regeneration. Taking a closer look at the top six targets for each TF in the SAM, we found that SBP23 regulates signal transduction and metabolism-related genes crucial for SAM development, whereas CUC2 and RLD1 target growth regulators and other TF involved in apical meristem identity, such as LEAFY (LFY), NON APICAL MERISTEM (NAM) and ROLLED LEAF 2 (RLD2). Interestingly, RLD1 also interacts with ARGONAUTE 1 (AGO1), supporting its role in the miRNA biogenesis pathway. Taking into account the important number of SBP23 and RLD1 targets in leaf, a relevant observation was their respective regulation by miR156 and miR166 at later regeneration stages (Figure 5).

## 3. Discussion

In vitro organogenesis is a unique process that evidences the pluripotency of somatic cells to form a whole plant. It exhibits how differentiated cells become stem cells again. Morpho-histological studies of the maize SE in several inbred lines revealed variable behaviors in dedifferentiated callus tissues and plant regeneration programs [5,28,32,33,34]. Recent reports employing Mexican Tuxpeño VS-535 are beginning to unravel the molecular features of its particular in vitro culture process [23,35]. Because the cultivar displays favorable agronomic characteristics, it is a desirable target for biotechnological purposes. Additionally, the knowledge generated on the molecular basis underlying successful plant regeneration and SE is greatly contributing to further application in other relevant maize cultivars.

Calli originated from different maize varieties display distinct features. Sun et al. described H99 inbred maize EC as yellow, friable and composed of small cells with dense cytoplasm, whereas the NEC, as spongy, watery and brown, with large cells separated by evident gaps [36,37]. Otherwise, the maize inbred line A19 EC and NEC were merely distinguished by the presence of meristematic cell clusters and parenchymal cells without a visible core, respectively [38]. Liu et al. reported the inbred line Y423 NEC as elongated cells without plastids, with larger vacuoles and intracellular spaces than EC [28]. For VS-535 we could reproducibly characterize the classical type II EC [27] and two different NEC tissues, Y-NEC and W-NEC, unable to produce plants under the established regeneration. NEC tissues displayed different proliferation patterns and presented tracheids, that could contribute to the system transport. The origin of EC is restricted to scutellar epithelial cells of the explant, supporting the idea that only certain parts of the embryo have regenerative capability [5,39].

During the staggered hormone depletion and photoperiod plant regeneration program for VS-535, many regenerative spots first develop in leaf structures before the root emergency. Their appearance on particular EC portion is still not well understood, but a plausible scenario involves an auxin maximum and the presence of stem cells derived from SAM. According to the literature, upon depletion of 2,4-D, auxin gradients restrict WUS expression to specific regions giving rise somatic embryos and shoot regeneration [40]. The morphological features described here, support plant regeneration taking place through a basipetal growth, that is, from the tip to the base of the leaf readily observed between the second and the third stages of development. Shoot growth becomes determined by the interplay between phytohormones, transcription factors and regulatory nodes operated by miRNAs [15,41]. On the other hand, plant root system results from the coordination between exogenous environmental factors and endogenous signal pathways. Possibly, the regenerative structure still adhered to the callus uses its conducting-like tissues (i.e., tracheids) to acquire nutrients from the culture medium. Hence, root development appears later, at a stage when the fully developed plantlet requires specific nutrient resources.

Although the role of miRNAs in SE has just started to arise, their involvement in almost any plant developmental process has been widely demonstrated [15]. Particularly, transcription factors and F-Box proteins targeted by miRNAs are crucial for cell division, spatial rearrangement, body patterning and auxin signaling, as well as during in vitro plant shoot regeneration and de novo SAM formation [18,25,42,43]. Maize miR156 controls the SBP-box transcription factor TASSELSHEAT4 (TSH4) levels for lateral meristem initiation and maintenance [44] and at least 19 maize SBPs corresponding to known *Arabidopsis* SPLs are potentially targeted by miR156 [45]. In addition, miR156 was recently identified as enhancer of the callus embryogenic potential in *Citrus sinensis* [46]. Furthermore, transcription factors CUC1 and CUC2 regulated by miR164 participate in the establishment and maintenance of axillary meristem and organ boundary during embryogenesis [47,48].

Another miRNA that functions as a short-range positional signal in *Arabidopsis* SAM to ensure stem cell identity is miR394 [42]. It down-regulates LCR, an F-box protein acting as inhibitor of the WUS downstream activation signal, to promote stem cell identity acquisition [43,49]. Recently, maize WUS2 has been demonstrated to strongly improve SE of recalcitrant lines [50]. At the same time, early WUS expression was described as marker for in vitro shoot organogenesis in tobacco and Indian spinach [51]. In Arabidopsis and maize, miR166 spatially restricts the localization of HD-ZIP III transcription factors at the shoot apex to establish the abaxial pattern in the emerging leaf [52]. This miRNA also participates as mobile signal for proper SAM maintenance and root development [42,53]. On the other hand, ARFs regulation by miR160 restricts stem cell niche and columella differentiation during root cap formation [54].

Here, we found that each one of these miRNAs display particular accumulation patterns, during plant regeneration (Figure 6A). While miR156, miR160 and miR166 accumulate at early stages, miR164 and miR394 are first reduced and further increase their levels upon leaf differentiation. We could not find the expected negative correlation between miR394 and its F-Box/LCR target possibly due to restricted spatial regulation as previously described (Figure 6B; [42]). For miR164 and CUC2 we found similar pattern, suggesting that the miRNA acts as a feedback loop to control the accumulation of its target in response to developmental switches. Also, while opposite correlation is expected between miRNA and target accumulation, due to preferential degradation promotion in plants [55], some reports have indicated no correlation for particular miRNA targets and additional regulation at the translational level, depending on the target [56].

Interestingly, all miRNA targets showed strikingly higher levels in the EC, whereas the WUS transcript was equally present in both EC and Y-NEC and at higher levels than in differentiated tissues (IE and during plant regeneration). This is in agreement with the previously reported miRNA abundance in established Tuxpeño VS-535 in vitro cultured calli [22], but also reveals important molecular differences between the two classes of proliferating tissues (EC and Y-NEC). As shown in Figure 6B, we propose that only the EC maintains the program of pluripotency using a battery of miRNAs and transcription factors able to operate within the nodes for SAM initiation and further root development in response to hormone depletion and light. During the first stage of plant regeneration a decrease of most targets was observed probably due to miRNA relocation as mobile signals to determine regenerative spots (Figure 6A). Particularly, the levels of CUC-2, RLD-1 and F-Box/LCR sharply decreased during plant regeneration, as expected from their spatial restriction within the emerging SAM. In turn, miR166 levels boosted at the 2nd developmental stage and further in the regenerated plantlet corresponding to its role in RLD1 regulation during leaf development. Also, miR160 displayed the highest levels at the 2nd stage coincident with the lowest ARF17 and ARF19 levels. ARF17 has been described as negative regulator of adventitious roots and its down-regulation by light and miR160 precedes root appearance [57]. At the 3^rd^ stage, and in the fully regenerated plantlet, miR160 accumulation decreased and ARF19 increased, supporting its distinct role during root formation. The expression pattern of these ARFs was similar to what reported for Arabidopsis SE [58] and maize zygotic embryogenesis [59]. However, further work is needed to dissect their role in the regeneration process.

According to our results and the model depicted in Figure 6, the VS-535 staggered plant regeneration process occurs in a basipetal way with miRNA-target most significant expression switches taking place at the 2nd and 3rd stages, in the absence of external hormones. It has been suggested that during the embryonic patterning in *Arabidopsis*, aerial structures are more dependent on miRNA production since mutants in *dicer-like 1* (*dcl1*) affected in all miRNA production display more striking phenotypes for SAM [17]. Hence, early miRNA function detected primarily for EC tissues prevents the over-accumulation of transcripts encoding differentiation-promoting factors [14]. Particularly, miR156-mediated repression of SPLs is required to avoid precocious expression of maturation phase genes, when active cell division program is still operating. At the same time, miR166/HD-ZIPIII pathway acts in parallel to repress the embryonic maturation programme [16]. Although more targets for these miRNAs should be analyzed in the future, the behavior of SBP23 and RLD1 nicely support their role in key state of differentiation for VS-535 SE program, including shoot regeneration, leaf and root development.

## 4. Material and Methods

### 4.1. Callus Induction and Subculture

Immature embryos from cultivar VS-535 were collected at 15–18 days after pollination of greenhouse-cultured plants. The ears were divided in portions of 6–8 cm, washed in sequential order with 70% ethanol, chlorine solution, and sterile deionized water. The embryos were carefully excised from each portion and placed in a solution of 0.25 g L^−1^ cefotaxime. Callus induction was performed on callus induction medium containing 2 mg L^−1^ 2,4-dichlorophenoxiacetic acid (2,4-D) as described previously [60]. Induced calli were subcultured on proliferation medium N6P, which included 0.3 mg L^−1^ 6-BA in addition to 2,4-D. Details for medium compositions are available in (Appendix A) Different calli types were identified and separated during the subsequent subcultures [60].

### 4.2. Plant Regeneration

Plant regeneration was carried out at different time points of callus subcultures. The procedure consisted of three culture medium changes under a light photoperiod (16 h light/8 h dark). Details for the method can be found in [60]. Briefly, the first change consisted of callus placement on N6P medium with half-reduction in hormone concentration (1 mg L^−1^ 2,4-D; 0.15 mg L^−1^ 6-BA). After two weeks, the calli were subcultured on N6P medium without hormones for two weeks (second change). For the third change, the regenerative structures were subcultured on Murashige and Skoog (MS) medium every two weeks until the formation of whole plantlets. The regeneration process was conducted in four different biological replicates to verify the morphology reproducibility for differentiation stages.

### 4.3. Morphological Characterization

The observations were made under a stereomicroscope (Olympus DF Plapo, Tokyo, Japan). Images were taken using a digital camera (Olympus U-cmad3, Miami, FL, USA) and edited with the Image-Pro Plus Software (Rockville, MD, USA). Callus induction was observed every week during a month, whereas the different types of calli and EC-derived tissues in plant regeneration were observed every two weeks to register the morphological changes taking place during differentiation.

### 4.4. Scanning Electron Microscopy

For scanning electron microscopy (SEM), samples from immature embryo (IE), IE on N6I medium upon seven days, callus types subcultured for 8 and 12 months and selected developmental stages during plant regeneration were fixed with FAA (10% formaldehyde, 5% acetic acid, 50% alcohol) for 24 h at room temperature. The tissue was washed three times in distilled water and dehydrated in a graded ethanol series (10, 30, 50, 70, 90 and 100%) for 24 h. The samples were dried by critical point drying with liquid carbon dioxide. Finally, the tissues were coated with gold for 2 min in a coater (Q150RES, Quorum, East Sussex, UK). The prepared samples were examined and photographed in a Hitachi SUI510 scanning electron microscope (Tokyo, Japan) operating at 10–15 kV.

### 4.5. Histological Analysis

For light microscopy, samples corresponding to different callus types and developmental stages through plant regeneration from two independent subculture times, were fixed on HistoChoice Tissue Fixative (Sigma–Aldrich, Saint Louis, MO, USA) according to the manufacturer’s instructions, dehydrated for 24 h through a graded series of ethanol at room temperature and embedded in paraplast at 60 °C, for 24 h. Longitudinal sections (10 µm) were cut and stained in safranin-fast green [61], the photos were taken with a *Motic BA210* trinocular compound microscope and edited with Motic Images Plus 2.0 Software (Motic, Wetzlar, Germany) and Jasc Paint Shop Pro (Jasc Software, Minneapolis, MN, USA).

### 4.6. RNA Isolation

RNA was isolated from immature embryo, EC, NEC and the four plant regeneration developmental stages in at least triplicate biological samples. Tissues used for all molecular analyses and the corresponding letter designations conserved throughout all figures are available in (Appendix A). RNA sized fractionation and clean-up were performed according to [60]. For the different plant regeneration developmental stages, callus surrounding regenerative tissues was kept at minimal. The RNA quality and concentration were determined using the NanoDrop Spectrophotometer (Thermo Scientific, Waltham, MA, USA).

### 4.7. qRT-PCR

For large RNAs, reverse transcription (RT) was performed using an oligo (dT) primer and the ImProm-II™ reverse transcription system (Promega, Madison WI, USA). Specific primers for each miRNA target were designed using Primer-BLAST optimized for real-time PCR (qPCR) and the amplification products were attempted to include the predicted miRNA-directed cleavage site. For small RNAs, the stem-loop RT and forward primers were designed according to [62]. All oligonucleotide sequences used in this study are available in (Appendix A).

Pulsed stem-loop RT was performed for all miRNAs and the U6 snRNA as control in duplicate reactions [60]. Measurements by qPCR were performed Maxima SYBR Green/ROX qPCR Master Mix in a 7500 Real-time PCR System (Applied Biosystems, Foster, CA, USA). Abundances were expressed as “fold change” using the 2^−∆∆*C*t^ method [63], considering the EC tissue as reference and rRNA 18S or U6 snRNA as internal housekeeping controls, respectively. Results were averaged for the biological and technical replicates (6–12 independent measurements). Statistical analyses were performed using one-way ANOVA method.

## 5. Conclusions

Different calli types with contrasting morphology, molecular features and regenerative potential were identified upon maize VS-535 dedifferentiation induced from 15 DAP-immature embryos. The embryogenic callus showed significantly higher levels of development-related miRNAs and their targets, whereas WUS2 levels were similar in embryogenic and non-embryogenic tissues. During plant regeneration, the function of miR156, miR160 and miR166 regulatory nodes correlated with early de novo SAM formation and reproducibly preceded root appearance. Furthermore, miR164 and miR394 transiently increased their levels during leaf development. According to this, we conclude that the initial miRNA and target levels in the callus underlie its regenerative potential and guide its ability to promote shoot appearance during plant regeneration from maize VS-535 in vitro cultures.

## Figures and Tables

**Figure 1 ijms-20-02079-f001:**
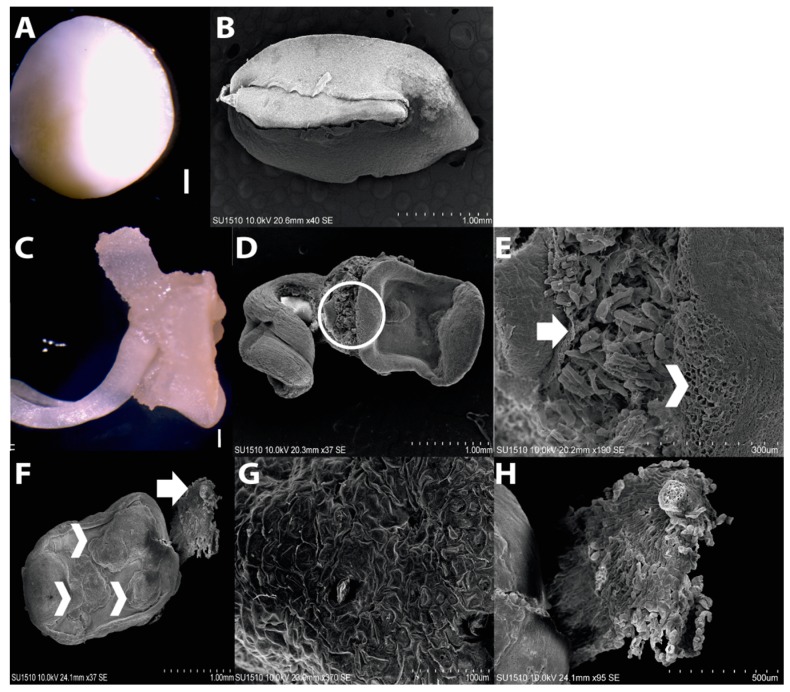
Origin of the different callus types for Tuxpeño VS-535. (**A**) Immature embryo (IE) at 15–18 days after pollination (DAP). (**B**) The IE observed with SEM. The embryonic axis and surrounding tissue are clearly distinguished. (**C**) The IE after seven days on N6I medium. Radicle protrusion and callus formation around the embryo are observed. (**D**) The IE after seven days on N6I medium observed with SEM. (**E**) Magnification of the circled area in **D**. The arrow points at non-embryogenic callus formation, whereas the arrowhead indicates pro-embryogenic callus tissue. (**F**) The IE after nine days on N6I medium observed with SEM. Cumulus of embryogenic masses formed on the scutellar epithelium are highlighted by several arrowheads, whereas an arrow points at non-embryogenic callus formed on the embryonic axis side. (**G**) Magnification of **F** in an arrowhead zone, (**H**) Magnification of **G** in the arrow zone.

**Figure 2 ijms-20-02079-f002:**
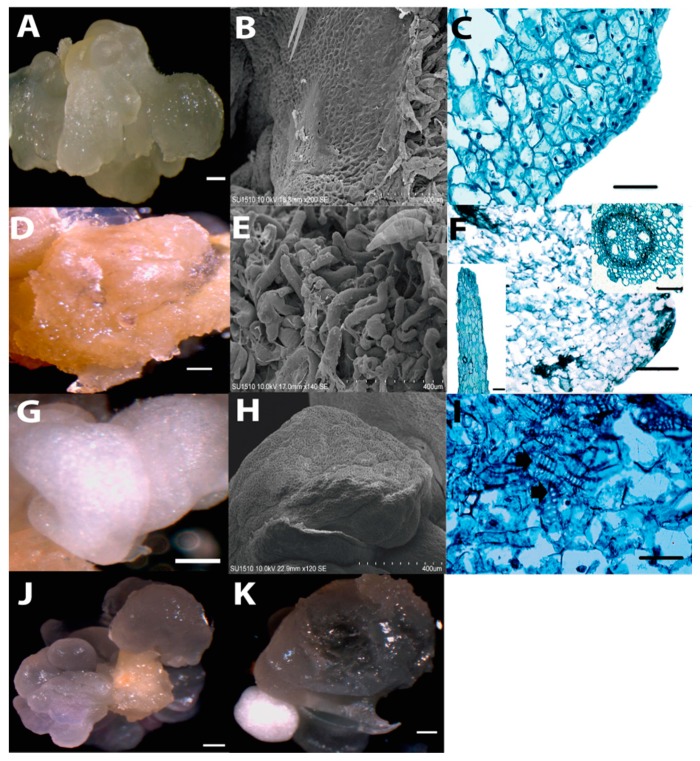
Different callus types observed during subcultures of Tuxpeño VS-535. (**A**) A portion of translucent embryogenic callus (EC). (**B**) EC observed with SEM. (**C**) Histological analysis of EC. Characteristic small isodiametric cells with prominent nuclei are observed at the callus surface. (**D**) A portion of yellow non-embryogenic callus (Y-NEC). (**E**) Y-NEC observed with SEM. (**F**) Histological analysis of Y-NEC. Larger cells with smaller nuclei can be appreciated. The inset images represent a root-like structure (right) and a vascular bundle (left) frequently observed for this callus type. (**G**) A portion of white non-embryogenic callus (W-NEC). (**H**) W-NEC observed with SEM. (**I**) Histological analysis of W-NEC. The arrows indicate tracheid structures present in this callus type. (**J**) and (**K**) Portions of callus tissues showing the co-existence of EC with Y-NEC and W-NEC, respectively. Histological analyses are shown at ×400 magnification. Bar in **A**, **D**, **G** and **H**: 1 mm. Bar in **C**, **F** and **I**: 100 μm.

**Figure 3 ijms-20-02079-f003:**
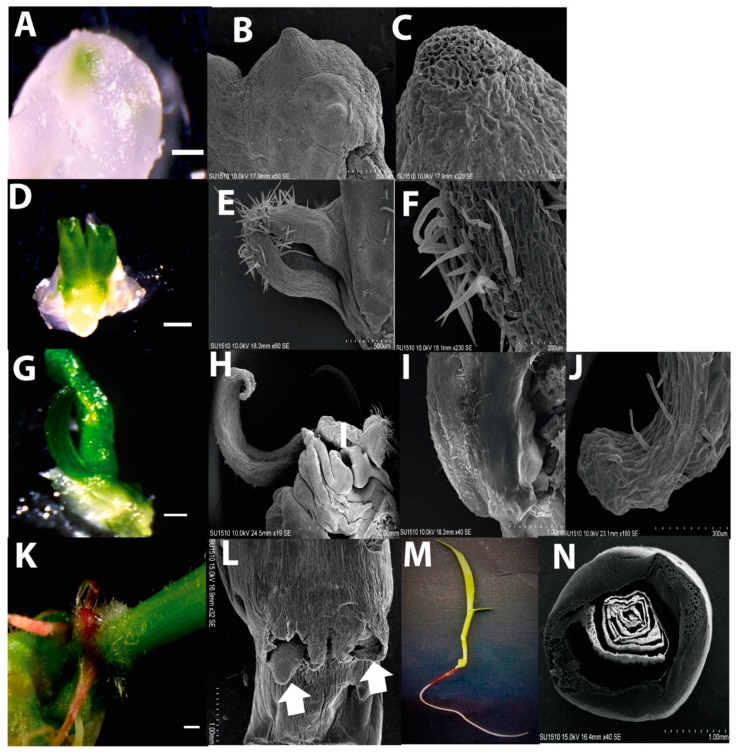
Developmental stages during Tuxpeño VS-535 plant regeneration from EC. (**A**) Regenerative spot observed on EC subcultured for two weeks in N6P with 50% hormones and photoperiod (1st stage). (**B**) The 1st stage of development observed with SEM. (**C**) Magnification of the regenerative spot tip with SEM. (**D**) Apical growth resembling leaf structure observed from a regenerative spot in N6P with 0% hormones (2nd stage). (**E**) The 2nd stage of development observed with SEM. (**F**) Magnification of the leaf-like structure exhibiting hairs the primordium. (**G**) Development of a differentiated leaf structure still adhered to the callus and without root emergence, in MS (3rd stage). (**H**) The 3rd stage of development observed with SEM. (**I**) Magnification of the basal portion of the leaf observed with SEM. (**J**) Magnification of the tip portion of the leaf observed with SEM. (**K**) The stem portion of a regenerated plantlet with visible roots grown in MS. (**L**) Roots emerging from the regenerated plantlet observed with SEM. Two arrows point at emerging roots. (**M**) Whole regenerated plantlet. (**N**) Cross section at the base of a plantlet stem observed with SEM. In **A**, **D**, **G** and **K** Bar= 1 mm.

**Figure 4 ijms-20-02079-f004:**
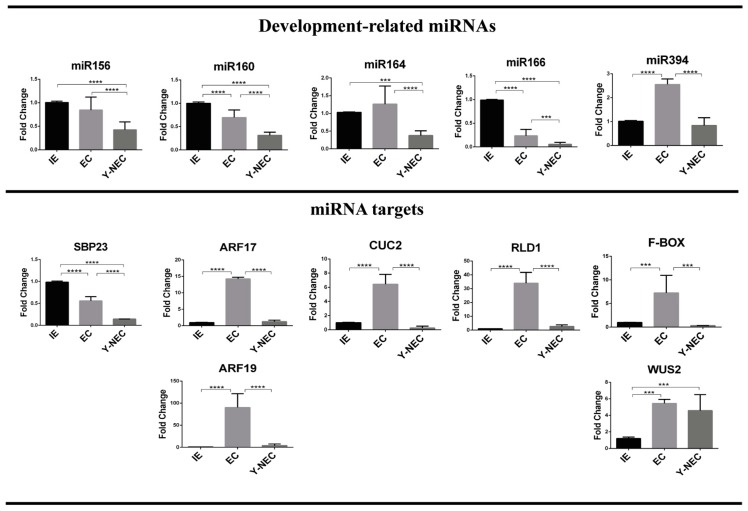
Development-related miRNAs and their target abundances in maize VS-535 SE. miRNA and mRNA abundances were analyzed by qRT-PCR in Tuxpeño VS-535 immature embryos (IE) used as explants to induce SE, and in established 12 month-old embryogenic (EC) and non-embryogenic (Y-NEC) callus tissues. Each miRNA target is shown below the corresponding miRNA, except for WUS2, which is not a direct target of miR394, but is instead regulated by an F-Box protein target of miR394. Fold change represents abundance relative to the IE and normalized by U6 snRNA internal control. Significant values are indicated as follows: (***) *p* < 0.001, (****) *p* < 0.0001.

**Figure 5 ijms-20-02079-f005:**
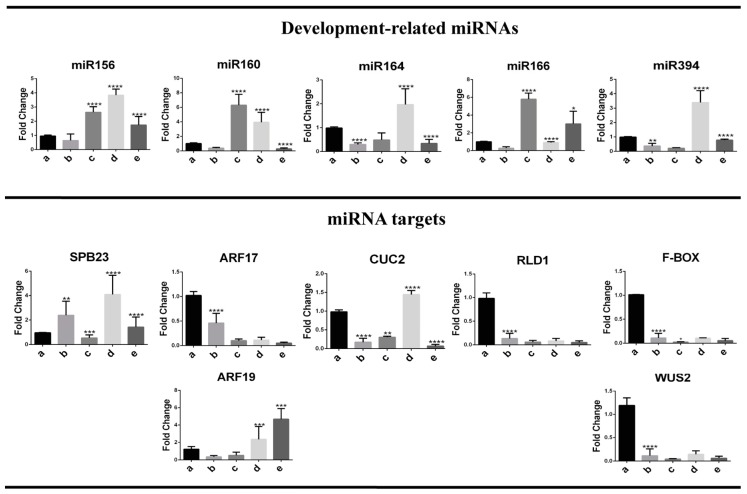
Abundance patterns of development-related miRNAs and their targets during maize VS-535 plant regeneration. miRNA and mRNA levels were analyzed by qRT-PCR during staggered hormone depletion and photoperiod for Tuxpeño VS-535 plant regeneration. (a) EC, (b) 1st stage of development, (c) 2nd stage of development, (d) 3rd stage of development, (e) Plantlet. Each miRNA target is shown below the corresponding miRNA, except for WUS2, which is not a direct target of miR394, but is instead regulated by an F-Box protein target of miR394. Fold change represents abundance relative to the EC and normalized by U6 snRNA internal control for miRNAs and 18S rRNA for targets. Significant values are indicated as follows: (*) *p* < 0.05, (**) *p* < 0.01, (***) *p* < 0.001, (****) *p* < 0.0001.

**Figure 6 ijms-20-02079-f006:**
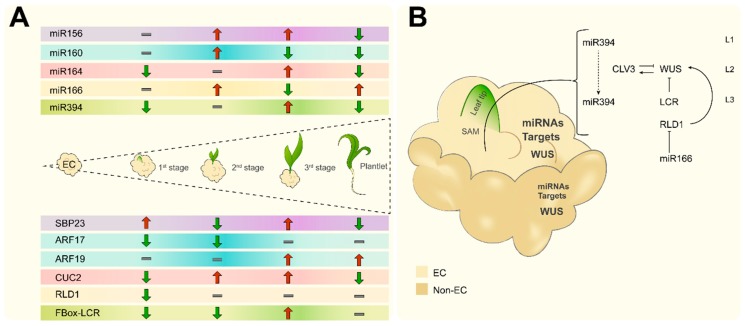
Proposed model for miRNA-target action during in vitro maize VS-535 Tuxpeño plant regeneration. (**A**) Maize VS-535 in vitro plant regeneration occurs in a basipetal way with the most noteworthy miRNA-target accumulation switches taking place at the 2nd and 3rd stages of regeneration. Opposite patterns between miRNA and targets are observed at these stages. (**B**) Distinctive miRNA-target accumulation between EC and Y-NEC (greater accumulation corresponds to bigger letter size) underlies the ability to develop regenerative structures under the appropriate stimulus. A highlighted node corresponds to miR394, which is proposed to relocate and negatively regulate LCR resulting in WUS spatial restriction and stem cell identity acquisition in SAM upon the regeneration stimulus. Likewise, miR166 participates as mobile signal for proper SAM maintenance and spatially restricts the localization of HD-ZIP III transcription factors at the shoot apex to establish polarity in the emerging leaf. Cell layers (L1–L3) and additional WUS regulation by CLAVATA (CLV3) are depicted according to the literature [42,43].

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
