# Peer review of "Development-Related miRNA Expression and Target Regulation during Staggered In Vitro Plant Regeneration of Tuxpeño VS-535 Maize Cultivar"

_ijms, 2019, doi:10.3390/ijms20092079_

Round 1
Reviewer 1 Report
Lopez-Ruiz and co-authors presented a study of miRNA-dependent regulation of target expression during in vitro regeneration of maize together with the morphological characterization of different callus types. The introduction is written concisely and briefly, however, gives the main information very clearly. Authors made a background for the research part in Introduction via touching the problem of in vitro regeneration of maize, genetic interactions between already known in other species TFs involved in shoot regeneration together with an insight into the miRNA-dependent regulatory role. The Introduction is finished with a clear aim of the study and the main conclusion has been given. In the Results section, I did not observe any problems until the fragment concerning miRNA and target regulation. Since the title is ‘Development-related miRNA expression and target regulation…” I’ve expected that most of the manuscript would be dedicated to that subject. I would recommend the Authors to re-check this paragraph and try to describe in a more detailed way the targets (i.e. what RLD1, SBP2, etc… is encoding and why is important for regeneration? – it would enrich the model proposed by Authors). Further, taking into account that there is a lot of genomic data for maize, it would be really interesting to check by in silico studies if there are any interactions between studied factors? It is possible to draw the interaction network that will highly enrich already interesting data obtained by Authors. Also, it should be possible to check if the described TFs bind to promoters of genes already known as involved in the development of embryogenic tissue. Together, these analyses would give an elegant supplement to the study.
As a minor comment: please check the resolution of graphs with expression, namely Fig 4 and 5, the font is really small and blurred.
Author Response
Reviewer 1:
In the Results section, I did not observe any problems until the fragment concerning miRNA and target regulation. Since the title is ‘Development-related miRNA expression and target regulation...” I’ve expected that most of the manuscript would be dedicated to that subject. I would recommend the Authors to re-check this paragraph and try to describe in a more detailed way the targets (i.e. what RLD1, SBP2, etc... is encoding and why is important for regeneration? – it would enrich the model proposed by Authors).
We have added a description on page 10, lines 303-308.
Further, taking into account that there is a lot of genomic data for maize, it would be really interesting to check by in silico studies if there are any interactions between studied factors? It is possible to draw the interaction network that will highly enrich already interesting data obtained by Authors. Also, it should be possible to check if the described TFs bind to promoters of genes already known as involved in the development of embryogenic tissue. Together, these analyses would give an elegant supplement to the study.
We added a regulatory gene network analysis using a platform generated for maize (Huang et al., 2018; BMC Plant Biology, 18:111; https://www.bio.fsu.edu/mcginnislab/mgrn/). According to the suggestion, we highlighted the relevance of targets for the miRNA-regulated TFs particularly in SAM development, which underlies our model. The information is provided as supplementary information, Figure S3 and presented in Results, page 10, lines 308-320.
As a minor comment: please check the resolution of graphs with expression, namely Fig 4 and 5, the font is really small and blurred.
We increased the letter size and resolution.
Reviewer 2 Report
Dear editor,
Thank you for giving me a chance to review the manuscript “ijms-484048, Development-related miRNA expression and target 2 regulation during staggered in vitro plant 3 regeneration of Tuxpeño VS-535 maize cultivar.
In this study, the authors described embryogenic callus (EC) formation from immature embryo (IE) explants of a maize cultivar (Tuxpeño VS-535). After optimizing plant tissue culture procedures, they analyzed expression levels of several miRNA genes (miR156, miR160, miR164, miR166 and miR394) and their target genes (SBP23, CUC1/2, ARF17/19, RLD1, F-BOX and WUS2) involved in regeneration and shoot apical meristem. They identified accumulation of miR164 and miR394 in EC. However, they did not observe reverse correlation between miRNA and target genes for all miRNAs. They also identified miRNAs associated with a particular stage of development e.g. miR394 accumulation at the 3rd stage.
This is an original study presenting EC and plantlet regeneration for maize IEs and regulation of expression of miRNA and regeneration-involved genes. As maize is monocot and can be regarded as tissue culture-recalcitrant, in vitro plant regeneration can be improved by targeting miRNAs and other regeneration-associated gene by siRNA methods. Therefore, this manuscript (ijms-484048) can be published in “International Journal of Molecular Sciences” as an original article. I have indicated several stylistic remarks on the manuscript.

Author Response
Reviewer 2:
I have indicated several stylistic remarks on the manuscript.
We have fulfilled all stylistic remarks and corrections provided in the pdf version of revised manuscript by this reviewer. According to these corrections, the order of references has been moved. We also deleted two references and added a new one.